# Development of an NT-ProBNP Assay Reagent Based on High Specific Activity Alkaline Phosphatase CmAP and an Improved Coupling Method

**Hai-Chao Li [1], Xin He [2], Shan-Peng Qiao [1], Zhen-Ni Liu [1] and Yu-Zhou Gao [1,\*]**

[1]  Department of Changchun Institute of Engineering Technology, Suzhou Institute of Biomedical Engineering and Technology, Chinese Academy of Science (CAS), Changchun 130052, China; lihc@sibet.ac.cn (H.-C.L.); qiaoshanpeng@163.com (S.-P.Q.); liuzn@sibet.ac.cn (Z.-N.L.)

[2]  Department of Jilin City Institute of Biological Products, Suzhou Institute of Biomedical Engineering and Technology, Chinese Academy of Science (CAS), Jilin 132013, China; Helix0804@163.com

\*  Correspondence: gaoyz@sibet.ac.cn; Tel.: +86-1-357-865-6691

**Abstract:** (1) Background: Chemiluminescent enzyme immunoassay (CLEIA) is an efficient analytical method. Alkaline phosphatase (ALP) with high specific activity is the basis for CLEIA to achieve high sensitivity. In this study, a high specific activity *Cobetia marina* ALP (CmAP) and an improved coupling method were used to develop an N-terminal pro-B-type natriuretic peptide (NT-proBNP) diagnostic reagent. (2) Methods: The purification method of CmAP was improved and the related enzyme activities were assessed. The enzyme and magnetic beads were coupled only to the Fc region of the detection antibody and the capture antibody, respectively, by using a specially improved method. The NT-proBNP in human serum was assessed. (3) Results: The specific activity of the purified CmAP was found to be 13,133 U/mg. No loss in the enzyme activity was observed after its storage at room temperature for 4 months. The sensitivity of the in vitro diagnostic reagents was found to be 0.58 ng/L. (4) Conclusions: CmAP can be applied as a substitute for the commercial ALP. Analytical parameters indicated that the chemiluminescence diagnostic reagent for NT-proBNP is adequately sensitive and reliable for detecting the serum NT-proBNP, which suggests that both the enzyme and coupling method are suitable for the CLEIA.

**Keywords:** chemiluminescence enzyme immunoassay (CLEIA); *Cobetia marina* alkaline phosphatase (CmAP); N-terminal prohormone of brain natriuretic peptide (NT-proBNP); antibody coupling

## 1. Introduction

To rapidly screen the biomarkers before routine clinical examination, the magnetic beads-based chemiluminescence immunoassay (MPs-CLEIA) technique has been used because of its high sensitivity and wide dynamic range, without the requirement for radioactive reagents [1–3]. The test sensitivity is closely related to the performance of the instrument and the paired antibody, and especially to the specific activity of the enzyme. The reaction of the enzyme-catalysed substrate is continuous and mild. When the amount of a chemiluminescence substrate is sufficient, the chemiluminescence signal increases with time under a certain time range. This process requires sufficient time for the reaction solution to mix evenly. Alkaline phosphatase (ALP), often used to label antibodies, can catalyse luminescent regents to produce an enhanced chemiluminescence after the competitive immunoreaction [4–6].

ALP (EC 3.1.3.1) catalyses the hydrolysis of phosphate compounds under alkaline conditions [7]. ALP exists in a wide range of organisms, from bacteria to humans. *Cobetia marina*, isolated from the coelomic liquid of a mussel, produces intracellular ALP (CmAP) that possesses high activity. The native

protein at the final purification stage has a specific activity of approximately 15,000 U/mg, which makes it one of the most active ALPs studied to date [8]. The *CmAP* gene was first cloned into *Escherichia coli*, and then the active protein was expressed successfully by Nasu et al. The purified recombinant CmAP also possesses a high specific activity of 12,700 U/mg [9].

Heart failure (HF) is considered a complex syndrome, and it is the final stage in the development of a coronary heart disease [10]. Early diagnosis and timely treatment are of great significance in improving the prognosis of patients with HF [11,12]. The N-terminal pro-B-type natriuretic peptide (NT-proBNP) can sensitively and specifically reflect changes in the left ventricular function [13–15]. B-type natriuretic peptide (BNP) is a preproBNP precursor, and it contains 134 amino acids. The precursor can be cleaved to a 108 amino acid polypeptide, which is called proBNP (precursor of BNP). The stimulation of proBNP through cardiomyocytes (e.g., cardiomyocyte stretching), results in the hydrolysis of proBNP into an NT-proBNP (1–76 amino acids) and a bioactive hormone BNP (77–108 amino acids) [16,17]. Both the peptides are released into the circulation. In plasma, the NT-proBNP and BNP molecules exist in a ratio of 1:1; the former is more stable than the latter, and can reflect changes in the ventricular function. The exclusion cut-off point of the NT-proBNP in acute HF is less than 250 ng/L [18].

On the basis of the available reports, we speculated that CmAP is suitable for antibody labelling. The Fc region of most antibodies contains an oligosaccharide chain, which is not present in the variable region. Based on this feature, we developed an improved antibody coupling method (coupling of enzymes to the detection antibody and that of magnetic bead to the capture antibody). In the present work, we developed a chemiluminescence in vitro diagnostic reagent by using CmAP and an improved coupling method. As already stated, the NT-proBNP is an important cardiac biomarker, and its structure is very simple. Therefore, in this study, we used NT-proBNP as a tool to develop a high sensitivity in vitro diagnostic reagent. Figure 1 illustrates the principle of CmAP-based MPs-CLEIA. The CmAP-labelled detection antibody and the magnetic bead-coupled capture antibody prepared in this study possess high antigen-binding efficiency. The analytical parameters were assessed to explore whether this CmAP-based chemiluminescence diagnostic reagent is sensitive and reliable. We hope that this research can promote the development of the CLEIA.

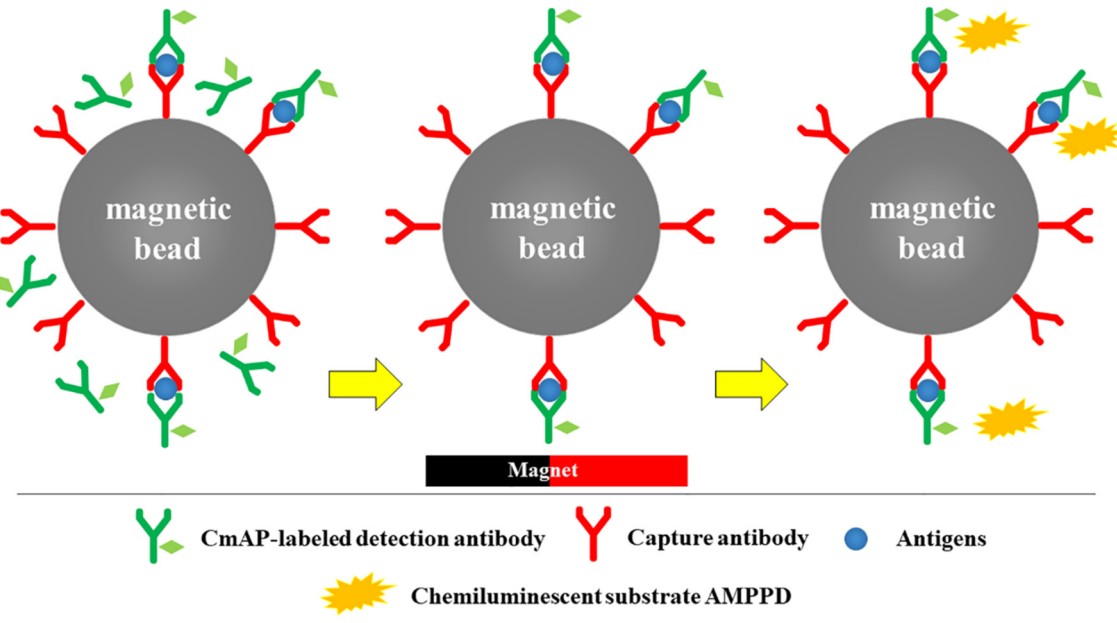

**Figure 1.** Schematic illustration of the magnetic beads-based chemiluminescence immunoassay (MPs-CLEIA).

## 2. Materials and Methods

### 2.1. Materials

Restriction enzymes and $T_4$ DNA ligase were purchased from TaKaRa (Dalian, China). Competent cells, plasmids, LB medium, ampicillin, isopropyl β-ᴅ-1-thiogalactoside (IPTG), bovine serum albumin (BSA), sodium periodate, cystamine dihydrochloride, sodium cyanoborohydride, dithiothreitol (DTT), succinimidyl-4-(N-maleimidomethyl)-cyclohexane-1-carboxylate (SMCC), N-ethylmaleimide (NEM), biotin-LC-hydrazides, and p-nitrophenyl phosphate (PNPP) were purchased from Sangon Biotech (Shanghai, China). The antibody was purchased from Fapon Biotech Inc. (Shenzhen, China). Streptomyces-coated immunomagnetic particles were purchased from Vdo Biotech Co., Ltd. (Suzhou, China). Blood samples were collected from The First Bethune Hospital of Jilin University (Changchun, China). The chemiluminescent substrate AMPPD (3-(2′-Spiroadamantane)-4-methoxy-4-(3″-phosphoryloxy) phenyl-1, 2-dioxetane), was purchased from Monobind Inc. (Lake Forest, CA, USA).

### 2.2. Instrumentation

All protein chromatography columns were procured from GE Healthcare (Uppsala, Sweden). High-speed refrigerated multifunction centrifuge was purchased from Beckman Coulter (Pasadena, CA, USA), and ultraviolet-visible spectrophotometer (UV-2700) was purchased from Shimadzu (Kyoto, Japan). The chemiluminescence analyser (LumiStation 1800) was purchased from Shanpu Biotechnology Co., Ltd. (Shanghai, China).

### 2.3. CmAP Cloning, Expression and Purification

The nucleotide sequence for the native *CmAP* gene was submitted to GenBank (accession number: ABD92772.1). We optimised a 1608-bp DNA sequence (including the N-terminal signal peptide sequence) for better protein expression in *E. coli*. The whole optimised gene was synthesised and inserted into a cloning vector PUC57 between the Nde I and BamH I restriction enzyme cutting sites by Sangon Biotech. The PUC57-*CmAP* and the expression vector pET16b were simultaneously digested using the restriction enzymes Nde I and BamH I. The digested product was ligated with $T_4$ DNA ligase after agarose gel purification. Finally, the recombinant plasmid was transformed into *E. coli* BL21 (DE3). The *CmAP* sequence was confirmed by Sangon Biotech (Shanghai, China).

The recombinant strains were cultured at 37 °C in 2 L of LB medium, with shaking at 180 rpm. When OD600 reached 0.8, a solution of IPTG was added into the culture to a final concentration of 0.5 mM. Then, the culture was allowed to grow overnight at 16 °C. The cells were harvested after centrifugation at $6000\times g$ for 10 min and suspended in buffer A (20 mM PBS, 0.05% $NaN_3$, and 0.5% Tween-20; pH 8.0). The suspended cells were lysed through ultrasonication and recentrifuged at $12,000\times g$ for 20 min to remove cell walls. The supernatant was loaded onto a High Q anion exchange column previously equilibrated with buffer A, and the enzyme was eluted under a concentration gradient of 0.1–1.0 M NaCl in buffer A. Afterwards, Phenyl HP chromatography, with a linear gradient of 1.0–0 M $(NH_4)_2SO_4$ in buffer A, was performed. The enzyme solution was finally concentrated and loaded onto a Sephacryl™ S-300 gel filtration column, and then eluted using buffer B (20 mM PBS, 0.05% $NaN_3$, 0.5% Tween-20, and 0.15 M NaCl; pH 8.0) at 1 mL/min. The fractions with enzymatic activity were desalted through dialysis after each step of chromatography. The purity of the protein was evaluated using sodium dodecyl sulphate-polyacrylamide gel electrophoresis (SDS-PAGE), and the protein concentration was determined through the Bradford method by using BSA as the standard. The purified enzyme was concentrated and stored at 4 °C.

### 2.4. Enzyme Activity Characterisation

The activity of the purified enzyme CmAP was assessed using the substrate PNPP [8]. The reaction was monitored at 405 nm ($\varepsilon_{405} = 18.5$ cm²/μmol) of the p-nitrophenol adsorption peak by using a

UV-2700 spectrophotometer. The reaction solution containing diethanolamine (DEA, 1 M; pH 10.3), 5 mM MgCl$_2$, and 15 mM PNPP was monitored for 5 min at 37 °C. One unit of the ALP activity was defined as the amount of the enzyme required to convert 1 μmol of p-nitrophenyl phosphate to p-nitrophenol in 1 min. Each measurement was taken three times.

## 2.5. ALP Labelling of the NT-proBNP Detection Antibody

The primary amino groups in the Fc region of the NT-proBNP detection antibody were modified by adding sulfhydryl groups before coupling of the antibody with CmAP. The detection antibody (0.5 mg) was dissolved in 100 μL of buffer 1 (0.1 M TEA; pH 8.0) containing 0.15 M sodium chloride and was placed in an amber vial. The detection antibody was incubated with 5 μmol sodium periodate at 4 °C for 1 h, desalted through gel filtration with a Sephadex G-25 column by using buffer 2 (0.1 M sodium phosphate and 0.1 M NaCl; pH 7.0), and concentrated to 100 μL. Cystamine dihydrochloride (20 μmol) and sodium cyanoborohydride (2 μmol) were added, and the solution was incubated overnight at room temperature. The cystamine derivatised antibody was desalted using a Sephadex G-25 column equilibrated with buffer 3 (0.1 M sodium phosphate and 0.1 M NaCl, and 2 mM EDTA; pH 7.0), and then concentrated to 100 μL as done previously. The concentrated product was treated with 2 mM DTT for 15 min at room temperature. The activated antibody containing thiol groups in the Fc region was desalted using a Sephadex G-25 column equilibrated with buffer 3.

The purified CmAP (0.6 mg) was activated by 30-fold molar excess of SMCC for 30 min at room temperature in buffer 4 (0.1 M sodium phosphate, 0.1 M NaCl, 1 mM MgCl$_2$, and 0.1 mM ZnCl$_2$; pH 7.0). After desalting ALP through the Sephadex G-25 column, the ALP was mixed with the activated detection antibody. The mixture was incubated at overnight at 4 °C. Unreacted thiol groups were capped with 0.5 mg NEM at room temperature for 2 h. The product was purified, concentrated, and stored at 4 °C with 500 μL of buffer 5 (0.1 M sodium phosphate, 0.05% NaN$_3$, 0.1% proclin, and 0.05% Tween-20; pH 7.6), and named CmAP-IgG.

## 2.6. Immunomagnetic Particles Labelling of the NT-proBNP Capture Antibody

The capture antibody (0.5 mg) was dissolved in 100 μL of buffer 1 and incubated with sodium periodate as done previously, Afterwards, it was desalted and concentrated to 100 μL. The antibody was mixed with biotin-LC-hydrazides (10 mM), and the mixture was incubated overnight at 4 °C. The excess biotin-LC-hydrazides was removed through a Sephadex G-25 column. The biotinylated antibody was allowed to react with 5 mg Streptomyces-coated immunomagnetic particles at room temperature for 2 h, and then washed thrice to remove the unbound antibodies. The product was stored at 4 °C with 500 μL of buffer 6 (0.1 M Tris-HCl, 0.05% NaN$_3$, 0.1% proclin, and 0.05% Tween-20; pH 7.4) and named MPs-IgG.

## 2.7. CLEIA Method

The traditional CLEIA method was used. 50 μL of diluted CmAP-IgG, MPs-IgG, and NT-proBNP sample were added into a microplate well, and the mixture was incubated at 37 °C for 20 min. The complex was formed in a sandwich way, as shown in Figure 1, then washed three times. Subsequently, 50 μL of assay buffer (4 M DEA; pH 10.3) and 150 μL of AMPPD were added, and the chemiluminescence signal was measured immediately after reaction in dark for 5 min. Each chemiluminescence assay was duplicated at least three times. Before the assay evaluation, we evaluated the effect of reagent dose on the chemiluminescence value. Specific contents of the in vitro assay included standard curve fitting, sensitivity, recovery, precision, linearity-dilution effect, specificity, interferences test, and correlation analysis using the Roche NT-proBNP diagnostic system.

## 3. Results

### 3.1. Characteristics of the Purified Recombinant CmAP

As described in Section 2.3, three purification steps, namely High Q anion exchange chromatography, Phenyl HP chromatography and Sephacryl$^{TM}$ S-300 gel filtration chromatography, were used. Each step was monitored by determining the enzyme activity (Table 1). The specific activity of the final enzyme was 13,133 U/mg. The SDS-PAGE result indicated that the molecular mass of one subunit is approximately 55 kDa. As shown in Figure S1, the molecular mass was higher than that of the purified ALP from *E. coli* (prepared in our own laboratory) but lower than that of ALP from bovine intestines (Roche, Basel, Switzerland).

**Table 1.** Purification of recombinant enzyme *Cobetia marina* ALP (CmAP) from *E. coli* BL21 (DE3).

| Method | Total Activity (U) | Total Protein (mg) | Specific Activity (U/mg) | Yield (%) | Purification (fold) |
|---|---|---|---|---|---|
| Crude extract | 61,300 | 472 | 130 | 100 | 1 |
| High Q | 44,180 | 49 | 902 | 72 | 7 |
| Phenyl HP | 33,800 | 5.2 | 6500 | 55 | 50 |
| Sephacryl$^{TM}$ S-300 | 31,520 | 2.4 | 13,133 | 51 | 101 |

Figure S2 shows the effect of pH on the activity of the purified enzyme. The maximum activity of 100% was achieved at pH 10.3. The relative activity was more than 90% when the pH values ranged between 9.5 and 10.5. When the pH was >10.7, the enzyme activity dropped sharply. Figure S3 presents the enzyme activity as a function of temperature at pH 10.3. The maximum activity of 100% was achieved at 45 °C. The enzyme activity increased with an increase in temperature when lower than 45 °C. As the temperature continued to rise, the enzyme activity gradually decreased, due to enzyme inactivation. Figure 2a presents the thermostability and storage stability of CmAP. The activity before incubation was defined as 100%. No significant change was observed after incubation for 1 h at 37 °C, and the relative activity was still more than 90% after treatment for 1 h at 45 °C. When the temperature was higher than 50 °C, the rate of enzyme inactivation gradually increased. The half-life time of ALP at 60 °C was approximately 25 min. As shown in Figure 2b, the enzyme activity did not change after storage for 12 months at 4 °C. The enzyme activity did not change after 4 months, when stored at 25 °C, but it began to decline from the fifth month. Thus, CmAP was proven to have perfect stability.

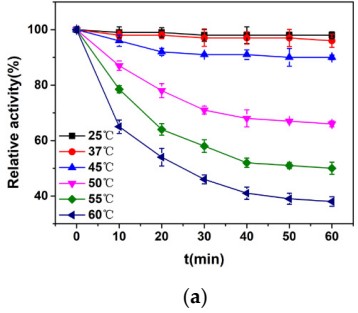
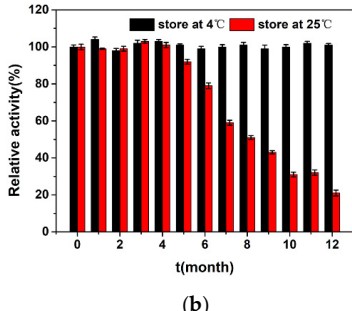

(a)          (b)

**Figure 2.** (**a**) Thermostability test of CmAP. 1 µg/mL CmAP solutions were incubated at temperatures ranging from 25 °C to 60 °C at pH 7.0 for 1 h. Then, the relative activity of each sample was measured at 37 °C, pH 10.3. The activity before incubation was defined as 100%. (**b**) Storage stability test of CmAP. CmAP solutions were stored at 4 °C and 25 °C for 12 months. Each sample was measured at 37 °C, pH 10.3. The activity before storage was defined as 100%.

### 3.2. Influence of Immunoreagents

The amount of immunoreagents is an important factor that affect the sensitivity and specificity of the CLEIA. Thus, the dosage of MPs-IgG and the dilution ratio of CmAP-IgG were studied and optimised. Before testing, CmAP-IgG was diluted in the following ratios: 1:20, 1:50, 1:100, 1:200, 1:300, 1:400, and 1:500. MPs-IgG was diluted to 0.1–0.8 mg/mL, according to the concentration of MPs. As shown in Figure 3, the optimum concentration of MPs was 0.3 mg/mL. Excessive MPs intensively affect the luminous intensity, because black MPs could unavoidably absorb the emitted light. Hence, 0.3 mg/mL MPs was used in the complete experiment. According to Figure 3, the chemiluminescence signal increased with a decrease in the dilution ratios of CmAP-IgG in the range of 1:20–1:500. As the dilution factor decreased to less than 1:20, the increase in the chemiluminescence intensity decreased, but the background value of chemiluminescence increased significantly (data not shown). Therefore, the dilution ratio of 1:100 of CmAP-IgG was used in subsequent tests in this study.

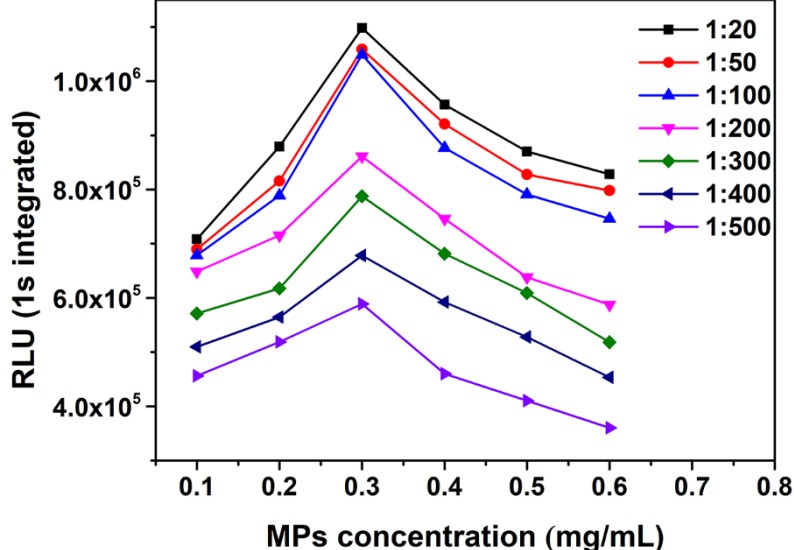

**Figure 3.** Influence of the dosage of MPs-IgG and the dilution ratio of CmAP-IgG studied using the standard N-terminal pro-B-type natriuretic peptide (NT-proBNP) concentration of 696 ng/L. The seven curves correspond to a series of dilution ratios of CmAP-IgG (i.e., 1:20, 1:50, 1:100, 1:200, 1:300, 1:400 and 1:500). All tests were performed at 37 °C.

### 3.3. Standard Curve and Sensitivity

After the dosages of CmAP-IgG and MPs-IgG were optimised, a series of NT-proBNP standard samples (0.0 ng/L, 5.7 ng/L, 40.7 ng/L, 65.7 ng/L, 136.0 ng/L, 696.0 ng/L, 2310.0 ng/L, 6450.0 ng/L, and 21,700.0 ng/L) were used. A standard curve was obtained through 4-parameter logistic curve fitting. The concentration of NT-proBNP was significantly correlated with the chemiluminescence value, as shown in Figure 4 ($R^2$ = 1). Twenty times of the blank sample were determined (Table S1), and then, the standard deviation was calculated. (SD = 289.6). The sensitivity (limit of detection (LoD)) of the assay was defined as 3SD/$K$ [19,20], where $K$ is the slope of the standard curve fit by the first 8 points, except the concentration of 21,700.0 ng/L ($K$ = 1489.1, $R^2$ = 0.999). Consequently, the sensitivity of this method was 0.58 ng/L. The value was sufficiently low for NT-proBNP diagnosis.

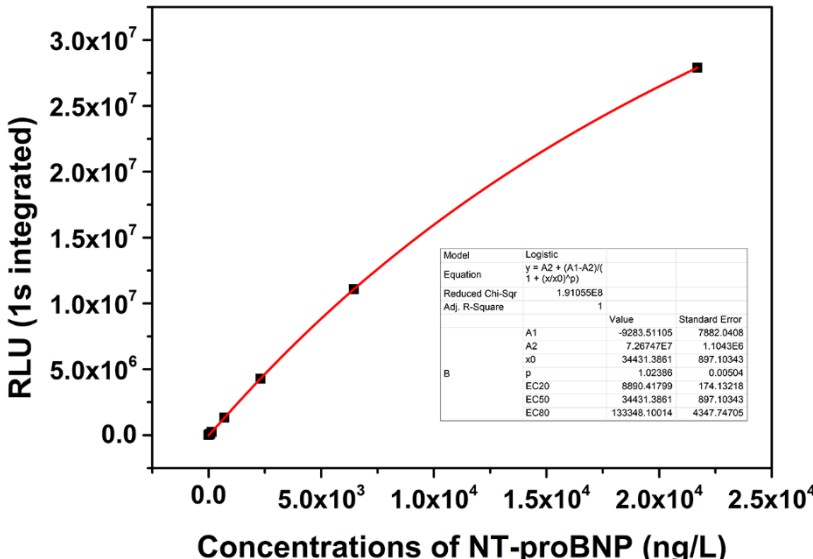

**Figure 4.** The relationship between chemiluminescence values and NT-proBNP concentrations by four parameter logistic curve fitting.

### 3.4. Recovery and Precision

Accuracy was studied through a recovery test. Different amounts of NT-proBNP were added to nine non-spiked human sera samples that included low, intermediate, and high values, and all tests were performed at least three times. As shown in Table S2, the results indicate that the recovery was between 95% and 105%. The precision of the test was determined by repeating the analysis of NT-proBNP samples. The coefficient of variation (CV) of intra-assay was determined using 10 replicates in one day. The CV of inter-assay was determined in 10 days by using the same method. As shown in Table S3, CV values were all less than 8% for the four analysed levels.

### 3.5. Linearity-Dilution Effect

The linearity-dilution effect was studied to select a standard human serum sample with a concentration of 2310.0 ng/L. It was diluted with horse serum and DEA buffer (1 M, pH 10.3) to obtain a series of concentrations (i.e., 1, 1/2, 1/8, 1/32, 1/64, and 1/128 of the original concentration). Each sample was measured in duplicate and in parallel. Data are shown in Tables S4 and S5. The linear regression equation was mapped with the expected value of NT-proBNP as the *X*-axis and the measured values of NT-proBNP as the *Y*-axis. The regression equations, $Y = 1.005X - 2.377$ ($R^2 = 0.9998$) and $Y = 0.9899X - 4.013$ ($R^2 = 0.9999$), were obtained when the serum sample was diluted with horse serum. Similarly, the regression equations, $Y = 0.9902X - 1.090$ ($R^2 = 0.9999$) and $Y = 1.005X + 3.647$ ($R^2 = 0.9998$), were obtained when the serum sample was diluted with DEA. The aforementioned results present a good correlation, and indicate that the matrix barely affects the binding of NT-proBNP to the antibody.

### 3.6. Specificity and Interferences

The specificity of the chemiluminescence diagnostic reagent for NT-proBNP was evaluated using other cardiac biomarkers that may affect the test results, with their concentrations being at least 10 times higher than their normal concentrations in the human body. As shown in Table 2, the selected pair of antibodies did not cross-react with the cTn complex, CK-MB, and myoglobin. The cross-reactivity with BNP was 0.0088%, which is absolutely acceptable in the analysis. The interference of bilirubin, haemoglobin, and triglyceride factors were assessed by overloading a serum sample (NT-proBNP = 546.0 ng/L). At the concentrations of 20 mg/dL, 500 mg/dL, and 3000 mg/dL of these respective factors, the interferences were all <10% in the NT-proBNP MPs-CLEIA assay (data not shown).

**Table 2.** Cross-reactivity of MPs-CLEIA with other cardiac biomarkers.

| Antigens | Tested Concentration (ng/mL) | NT-proBNP Concentration Determined (ng/L) | CR (%) |
|---|---|---|---|
| cTn complex | 100 | NA | 0 |
| CK-MB | 100 | NA | 0 |
| Myoglobin | 500 | NA | 0 |
| BNP | 100 | 8.8 | 0.0088 |

NA: not detected.

### 3.7. Comparison with the Roche NT-proBNP Diagnostic System

The results obtained using the proposed chemiluminescence diagnostic reagent in the determination of NT-proBNP in 100 clinical sera samples were compared with those obtained using the Roche Cobas e411 NT-proBNP diagnostic system. As shown in Figure 5, the $X$-axis is the result of Roche's system ($c_0$, ng/L), and the $Y$-axis is the detection result of this study ($c_1$, ng/L). The correlation coefficient (r) was 0.9851, which indicates a good agreement between the two diagnostic systems.

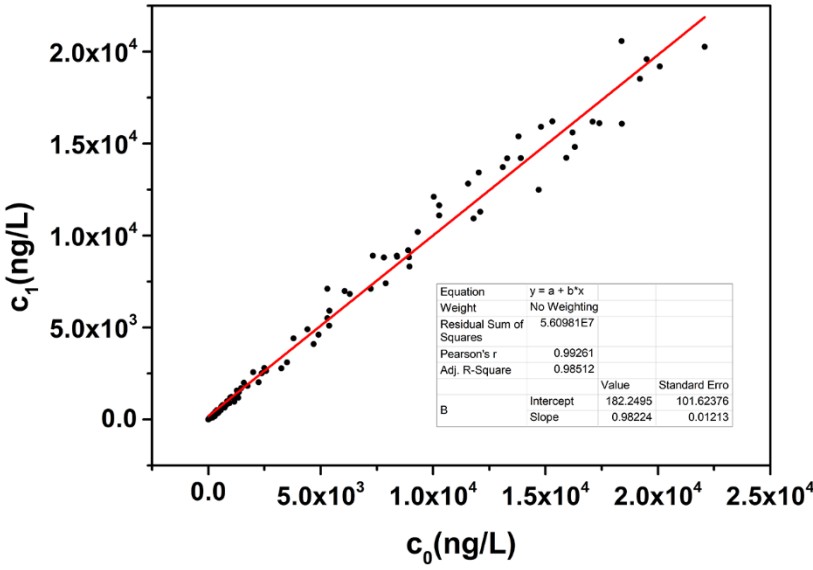

**Figure 5.** Correlation between results measured using the commercially available Roche NT-proBNP diagnostic kit ($c_0$) and the proposed chemiluminescence diagnostic reagent ($c_1$). Y = 0.9322X + 182.2 (r = 0.985, $p < 0.001$).

## 4. Discussion

CLEIA is one of the most commonly used in vitro diagnostic methods that possesses the longest chemiluminescence time. However, CLEIA also has some limitations. For example, the sensitivity of the test is over dependent on the specific activity of the enzyme, however, enzymes are generally proteins and are easily inactivated. Therefore, specific activity and stability of the marker enzyme are highly required in the CLEIA. CmAP is derived from a deep-sea bacterium that can adapt to extreme environments and possesses the highest specific activity of up to 15,000 U/mg. According to the method of Nasu et al. [9], we expressed CmAP in *E. coli* and improved the purification method. Finally, we obtained the purified enzyme with specific activity of more than 13,000 U/mg. Its specific activity is much higher than that of all commercial ALPs. The analysis of enzyme properties indicated that the enzyme possesses extremely strong stability, and almost no loss in its activity was observed when it was stored at room temperature for 4 months. The aforementioned advantages indicate that CmAP can be used as an alternative to commercial ALP as a marker enzyme.

Normally, the Fc region of most antibodies has a conserved oligosaccharide chain, but the Fab region lacks this feature [21]. In this study, this characteristic feature of an antibody was exploited. In the preparation of enzyme-labelled antibody, the oligosaccharide chain in the Fc region was oxidised to the aldehyde group through a reaction with sodium periodate, and then the reactive thiol group was produced in the Fc region through the aldehyde amine condensation reaction. The thiol group was coupled with SMCC-treated CmAP, so that CmAP could specifically bind to the Fc region of the antibody without affecting the variable region of the antibody. At present, Traut's reagent is being used in most studies to introduce sulfhydryl groups onto the antibody surface [19,22], but this reaction is random. Similar to the detection antibody coupling method, we developed a new method by coupling magnetic beads to the capture antibody. We used biotin-LC-hydrazide to react with the aldehyde group formed by the oligosaccharide chain oxidation to form a stable hydrazone bond, and then connected it with streptavidin magnetic beads through biotin at the other end, which also did not affect the variable region of the capture antibody. However, until now, most studies have used flourescein isothiocyanate to modify the capture antibody, which could also randomly react with amino groups on the surface of the antibody [19,23–25]. The modified group may occupy the variable region of the capture antibody. The NT-proBNP chemiluminescence diagnostic reagent was prepared using the improved method described in this paper. Compared with the traditional coupling method, the improved coupling method leads to better performance of the diagnostic reagent (data not shown).

The sensitivity of CLEIA to NT-proBNP was 0.58 ng/L in this work. This is the lowest sensitivity value of CLEIA in NT-proBNP detection until now, which can be used for the diagnosis of acute HF. However, the sensitivity of the electrochemiluminescence immunoassay (ECLIA) to NT-proBNP can reach or be even lower than 0.1 ng/L [20,26], which is much lower than that of CLEIA. Although extremely high sensitivity is the most important advantage of ECLIA, the cost of using and maintaining the instrument is very high. Sometimes, clinical diagnosis does not need such a high sensitivity, and therefore, it is not a cost-effective technique. CLEIA, by contrast, is much cheaper and more user-friendly. The chemiluminescence reaction of CLEIA is mild and persistent, which is convenient for data acquisition. CLEIA is still a very common chemiluminescence in vitro diagnostic method at present because of its unique advantages. The recovery and precision test results suggest that the diagnostic reagent has good accuracy and repeatability. The CV value of inter-assay is less than 8%, which indicate that the antibody coupling method used in this study is reliable. In addition to NT-proBNP, some antigens, such as cTn complex, CK-MB, myoglobin, and BNP, are also common cardiac biomarkers. Linearity-dilution effect results revealed that the matrix barely affects the binding of NT-proBNP to the antibody. Before the enzyme reacts with the substrate AMPPD, it needs to be washed three times, so that the matrix does not affect the catalytic reaction of the enzyme. According to the specificity assay, traces of cross-reactivity with BNP have been found. The number of moles of BNP and NT-proBNP secreted in the human body is the same, and the stability of the former is far lower than that of the latter, and thus, the cross-reactivity is negligible. Moreover, the good correlation with the Roche NT-proBNP diagnostic system indicates that our in vitro diagnostic reagent is valuable.

## 5. Conclusions

In this study, a chemiluminescence in vitro diagnostic reagent for NT-proBNP was synthesised by using CmAP with high specific activity and an improved coupling method. The high specific activity and high stability of CmAP indicate that CmAP is suitable as a chemiluminescence labelling enzyme. If large-scale industrial production is desired, CmAP can be used as a substitute for commercial ALP. The analytical parameters demonstrated that the chemiluminescence diagnostic reagent for NT-proBNP is sensitive and reliable for detecting serum NT-proBNP. NT-proBNP is a small molecule antigen with stable property and simple structure. It is suitable for methodology validation, as described in this study. Although such a high sensitivity may not be required in the clinical diagnosis of NT-proBNP, this study will serve as a useful reference for the development of in vitro diagnostic reagents for other complex antigens.

**Supplementary Materials:** The following are available online at http://www.mdpi.com/2076-3417/10/23/8682/s1, Figure S1: Coomassie-blue stained SDS-PAGE gel profile (12%) of different purified alkaline phosphatase, Figure S2: The effect of pH on the relative activity of purified CmAP, Figure S3: The effect of temperature on the relative activity of purified CmAP, Table S1: The chemiluminescence values of blank sample, Table S2: The recovery rate of NT-proBNP tested in normal human serum, Table S3: The CV values of intra-assay and inter-assay. Table S4: Linearity-dilution effect (diluted by horse serum), Table S5: Linearity-dilution effect (diluted by DEA buffer).

**Author Contributions:** H.-C.L. and Y.-Z.G. contributed to the conception and design of the study, performed the experiments, analysed the data and wrote the manuscript. X.H., S.-P.Q. and Z.-N.L. performed the experiments, analysed the data. All authors have read and agreed to the published version of the manuscript.

**Funding:** This study was supported by grants from Science and Technology Development Planning of Jilin Province (Grant No. 20180201060YY), and Science and Technology Development Planning of Jilin Province (Grant No. 20190303110SF).

**Conflicts of Interest:** The authors declare no conflict of interest.

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
