# Peer review of "Development of an NT-ProBNP Assay Reagent Based on High Specific Activity Alkaline Phosphatase CmAP and an Improved Coupling Method"

_applsci, doi:10.3390/app10238682_

Round 1
Reviewer 1 Report
This is a well conducted and well described investigation of the usefulness of a highly purified alkaline phosphatase as a reagent in the chemiluminescent assay for measurement of NT-proBNP in human serum. The quality of English writing is good, but it could be improved with input from a well-versed writer of English.
Author Response
Reviewer: 1
Comments: This is a well conducted and well described investigation of the usefulness of a highly purified alkaline phosphatase as a reagent in the chemiluminescent assay for measurement of NT-proBNP in human serum. The quality of English writing is good, but it could be improved with input from a well-versed writer of English.
Response:
Thank you very much for your careful review of our manuscript. The revisions you suggested required a thorough effort on our part. We believe that the manuscript has substantially improved paying attention to your suggestions. English expression has been carefully improved throughout the manuscript. We have used “Track Changes” function in the revision.

Reviewer 2 Report
In the paper, the authors have used alkaline phosphatase from Cobetia marina (CmAP) for developing a very sensitive chemiluminometric assay for NT-proBNP, a biomarker that is used clinically to assess patients who are suspected of having cardiac failure. The authors have purified the enzyme, coupled it to detection antibodies, and employed a sandwich assay with beads as solid phase for measuring NT-proBNP. The enzyme’s and the resulting assays characteristics have been investigated.
Major issues
1. The assay development, and its various evaluations and validations are scientifically sound. However, it is unclear as to why an ultrasensitive NT-proBNP assay, with such a low detection limit would be required in clinical practice. Current NT-proBNP assays, many of which are automated, are fit for purpose and have detection limits that are adequately low, not only for the cardiology setting in which they are used, but also for other potential scenarios, such as investigation of hyponatraemia. (Serum NT-proBNP thresholds for considering heart failure are at least two- or three-fold the detection limit of current automated assays.) Therefore, unless the authors can describe clinical situations in which there is a requirement for a very sensitive NT-proBNP assay, the emphasis and merit of the work in the way that it has been presented becomes rather dubious. In the light of the novelty of the use of CmAP, the authors may therefore wish to modify their paper to describe the work as a proof-of-principle (using NT-proBNP as an example) for the use of CmAP in EIAs for developing super-sensitivity assay with a wide dynamic range.
2. The use of English in the paper is quite poor, and most of the text would need to be re-written or improved, and reviewed by a native English speaker.
Other remarks
As it stands, and in relation to the point 1 above, how the increased sensitivity of the developed assay would help decision-making in clinical practice should be outlined briefly.
The Method should have a section that gives a brief overview of the work that was carried out for the assay evaluation/validation. These have been described in the Results section instead of the Methods.
Lines 214-5: The K in 3SD/K should be defined, i.e. the slope of the calibration curve.
Section 2.7: The authors should describe what they used as the assay buffer.
Section 3.6: The dilution studies were carried out using horse serum. Although in clinical practice there is no need to dilute samples with very high NT-proBNP, do authors have data comparing the results of dilutions in assay buffer with those in horse serum in order to obtain an indication of the degree of matrix effect?
Lines 233-4: cTn complex, CKMB and myoglobin are not structurally related to NT-proBNP.
Discussion: The authors have not discussed their results in terms of their assay evaluations well, and have focused mostly on the enzyme and the coupling procedures in their discussion.
Author Response
Reviewer: 2
Comments:
In the paper, the authors have used alkaline phosphatase from Cobetia marina (CmAP) for developing a very sensitive chemiluminometric assay for NT-proBNP, a biomarker that is used clinically to assess patients who are suspected of having cardiac failure. The authors have purified the enzyme, coupled it to detection antibodies, and employed a sandwich assay with beads as solid phase for measuring NT-proBNP. The enzyme’s and the resulting assays characteristics have been investigated.
Response:
Thank you very much for your careful review of our manuscript. The revisions you suggested required a thorough effort on our part. We believe that the manuscript has substantially improved paying attention to your suggestions.
Major issues
- The assay development, and its various evaluations and validations are scientifically sound. However, it is unclear as to why an ultrasensitive NT-proBNP assay, with such a low detection limit would be required in clinical practice. Current NT-proBNP assays, many of which are automated, are fit for purpose and have detection limits that are adequately low, not only for the cardiology setting in which they are used, but also for other potential scenarios, such as investigation of hyponatraemia. (Serum NT-proBNP thresholds for considering heart failure are at least two- or three-fold the detection limit of current automated assays.) Therefore, unless the authors can describe clinical situations in which there is a requirement for a very sensitive NT-proBNP assay, the emphasis and merit of the work in the way that it has been presented becomes rather dubious. In the light of the novelty of the use of CmAP, the authors may therefore wish to modify their paper to describe the work as a proof-of-principle (using NT-proBNP as an example) for the use of CmAP in EIAs for developing super-sensitivity assay with a wide dynamic range.
Response:
Thank you very much for your comments and advice. We have modified the title, abstract, introduction, and conclusion of the paper to a great extent for this issue. The innovation points of the research are to develop chemiluminescence in vitro diagnostic reagents by using high specific activity alkaline phosphatase CmAP and an improved coupling method. Taking NT-proBNP as an example, NT-proBNP not only has a very important clinical significance, but also has the advantages of simple structure, stable nature and strong anti-interference ability. The experimental results are beyond the expected standard, which proves that our assumption can provide reference for the development of in vitro diagnostic reagents for other complex structures antigens. In fact, our ultimate goal is to develop chemiluminescence diagnostic reagents for cTnI. Beckman Coulter measured plasma cTnI concentration of heparin lithium in 1089 apparently healthy people, and calculated the 99th percentile average value of 17.5 ng/L, which was far less than the cut-off value of NT-proBNP. The structure of cTnI is very complex, and it can form complex with cTnT and cTnC in plasma, there are very high requirements for antibodies. As a matter of fact, we have not yet screened out the appropriate matching antibody, which leads to the inter-assay difference of more than 10%, which needs further study. Therefore, we did not mention the cTnI related research in the paper. The major changes are as follows (marked in highlight in the text):
Title: Development of an NT-proBNP diagnostic reagent based on high specific activity alkaline phosphatase CmAP and an improved coupling method
Abstract: “In this study, a high specific activity Cobetia marina ALP (CmAP) and an improved coupling method were used to develop an N-terminal pro-B-type natriuretic peptide (NT-proBNP) diagnostic reagent.”(line 16-18); “which suggests that both the enzyme and coupling method are suitable for the CLEIA.”(line 26-27)
Introduction: Rewrote the last paragraph. “On the basis of the available reports, we speculated that CmAP is suitable for antibody labelling. The Fc region of most antibodies contains an oligosaccharide chain, which is not present in the variable region. Based on this feature, we developed an improved antibody coupling method (coupling of enzymes to the detection antibody and that of magnetic bead to the capture antibody). In the present work, we developed a chemiluminescence in vitro diagnostic reagent by using CmAP and an improved coupling method. As already stated, the NT-proBNP is an important cardiac biomarker, and its structure is very simple. Therefore, in this study, we used NT-proBNP as a tool to develop a high sensitivity in vitro diagnostic reagent. Figure 1 illustrates the principle of CmAP-based MPs-CLEIA. The CmAP-labelled detection antibody and the magnetic bead-coupled capture antibody prepared in this study possess high antigen-binding efficiency. The analytical parameters were assessed to explore whether this CmAP-based chemiluminescence diagnostic reagent is sensitive and reliable. We hope that this research can promote the development of the CLEIA.”(line 62-73)
Conclusions: “NT-proBNP is a small molecule antigen with stable property and simple structure. It is suitable for methodology validation, as described in this study. Although such a high sensitivity may not be required in the clinical diagnosis of NT-proBNP, this study will serve as a useful reference for the development of in vitro diagnostic reagents for other complex antigens.”(line 329-333)
- The use of English in the paper is quite poor, and most of the text would need to be re-written or improved, and reviewed by a native English speaker.
Response:
Thank you for your advice, English expression has been carefully improved throughout the manuscript. We have used “Track Changes” function in the revision.
Other remarks
- As it stands, and in relation to the point 1 above, how the increased sensitivity of the developed assay would help decision-making in clinical practice should be outlined briefly.
Response:
Thank you for your comments. We have made the following improvements in the conclusion part according to your suggestion: “NT-proBNP is a small molecule antigen with stable property and simple structure. It is suitable for methodology validation, as described in this study. Although such a high sensitivity may not be required in the clinical diagnosis of NT-proBNP, this study will serve as a useful reference for the development of in vitro diagnostic reagents for other complex antigens.”(line 329-333)
- The Method should have a section that gives a brief overview of the work that was carried out for the assay evaluation/validation. These have been described in the Results section instead of the Methods.
Response:
Thank you for your comments. We have given a brief overview of the analysis assay in section 2.7: “Before the assay evaluation, we evaluated the effect of reagent dose on the chemiluminescence value. Specific contents of the in vitro assay included standard curve fitting, sensitivity, recovery, precision, linearity-dilution effect, specificity, interferences test and correlation analysis using the Roche NT-proBNP diagnostic system.”(line 162-165)
- Lines 214-5: The Kin 3SD/Kshould be defined, i.e. the slope of the calibration curve.
Response:
Thank you for your comments. In this paper, we used 4-parameter logistic curve fitting method to draw the standard curve, which was to consider the test of higher concentration. Now K is the slope of the standard curve fit by the first 8 points except the concentration of 21700.0 ng/L (K=1489.1, R2=0.999). These eight points are almost in a straight line. As a result, the sensitivity by using this method is 0.58 ng/L. The original Section 3.3 and 3.4 were merged into Section 3.3. The major changes are as follows: “Twenty times of the blank sample were determined (Table S1), and then, the standard deviation was calculated. (SD = 289.6). The sensitivity (limit of detection, LoD) of the assay was defined as 3SD/K [19,20], where K is the slope of the standard curve fit by the first 8 points except the concentration of 21700.0 ng/L (K = 1489.1, R2 = 0.999). Consequently, the sensitivity of this method was 0.58 ng/L. The value was sufficiently low for NT-proBNP diagnosis.”(line 218-223)
- Section 2.7: The authors should describe what they used as the assay buffer.
Response:
Thank you for your advice. We have made the following changes to Section 2.7: “Subsequently, 50 μL of assay buffer (4 M DEA; pH 10.3) and 150 μL of AMPPD were added”(line 160)
- 7. Section 3.6: The dilution studies were carried out using horse serum. Although in clinical practice there is no need to dilute samples with very high NT-proBNP, do authors have data comparing the results of dilutions in assay buffer with those in horse serum in order to obtain an indication of the degree of matrix effect?
Response:
Thank you for your comments. We redesigned and implemented the experiment according to your suggestion. The data is presented in Tables S4, and S5 of Supplementary Materials. We have re-fitted the data linearly. The results shown in Section 3.5 demostrate that the matrix can hardly affected the binding of NT-proBNP to antibody. Before the enzyme reacts with the substrate, it needs to be washed thoroughly three times. As long as the binding between antigen and antibody is not affected, the final result will not be affected. The detailed changes we made in our manuscript are as follows: “The linearity-dilution effect was studied to select a standard human serum sample with a concentration of 2310.0 ng/L.”(line 237-238); “The regression equations, Y = 1.005X – 2.377 (R2 = 0.9998) and Y = 0.9899X – 4.013 (R2 = 0.9999), were obtained when the serum sample was diluted with horse serum. Similarly, the regression equations, Y = 0.9902X - 1.090 (R2 = 0.9999) and Y = 1.005X + 3.647 (R2 = 0.9998), were obtained when the serum sample was diluted with DEA. The aforementioned results present a good correlation and indicate that the matrix barely affects the binding of NT-proBNP to the antibody.”(line 242-246)
- Lines 233-4: cTn complex, CKMB and myoglobin are not structurally related to NT-proBNP.
Response:
Thank you for pointing out our mistake. We have changed the presentation in Section 3.6: “The specificity of the chemiluminescence diagnostic reagent for NT-proBNP was evaluated using other cardiac biomarkers that may affect the test results”(line 248-249)
- Discussion: The authors have not discussed their results in terms of their assay evaluations well, and have focused mostly on the enzyme and the coupling procedures in their discussion.
Response:
Thank you for your comments. We have improved on the last paragraph of the discussion section. Now the impact of the analysis results was evaluated: “The recovery and precision test results suggest that the diagnostic reagent has good accuracy and repeatability. The CV value of inter-assay is less than 8%, which indicate that the antibody coupling method used in this study is reliable. In addition to NT-proBNP, some antigens such as cTn complex, CK-MB, myoglobin and BNP are also common cardiac biomarkers. Linearity-dilution effect results revealed that the matrix barely affects the binding of NT-proBNP to the antibody. Before the enzyme reacts with the substrate AMPPD, it needs to be washed three times, so that the matrix does not affect the catalytic reaction of the enzyme. According to the specificity assay, traces of cross-reactivity with BNP have been found. The number of moles of BNP and NT-proBNP secreted in the human body is the same, and the stability of the former is far lower than that of the latter, and thus, the cross-reactivity is negligible. Moreover, the good correlation with the Roche NT-proBNP diagnostic system indicates that our in vitro diagnostic reagent is valuable.”(line 311-322)

Reviewer 3 Report
In this paper, Li et al. report the results of the development of a high-performance chemiluminescence enzyme immunoassay for N-terminal pro-B-type natriuretic peptide (NT-proBNP). The analysis seems accurate and without major flaws. Please note that NT-proBNP should be better expressed as ng/L.
Author Response
Reviewer: 3
Comments:
In this paper, Li et al. report the results of the development of a high-performance chemiluminescence enzyme immunoassay for N-terminal pro-B-type natriuretic peptide (NT-proBNP). The analysis seems accurate and without major flaws. Please note that NT-proBNP should be better expressed as ng/L.
Response:
Thank you very much for your suggestions. Now we have changed all “pg/mL” to “ng/L” throughout the manuscript.

Round 2
Reviewer 2 Report
This paper has been improved significantly. The emphasis is now more on the novel assay reagent per se, with NT-proBNP assay development used as a proof of principle. The English usage has been improved significantly, and the other issues that had been raised before have been addressed satisfactorily.
In the Title, ‘diagnostic’ should be replaced with ‘assay’.
With respect to the emphasis of the paper that is now on the rather novel assay reagent, would it be better if the text on NT-proBNP (or preferably a shortened version of it) with which the Introduction starts (lines 33-44) was incorporated into the last paragraph of the Introduction?
Author Response
Reviewer: 2
Comments:
This paper has been improved significantly. The emphasis is now more on the novel assay reagent per se, with NT-proBNP assay development used as a proof of principle. The English usage has been improved significantly, and the other issues that had been raised before have been addressed satisfactorily.
Response:
Thank you very much for your careful review of our manuscript. The revisions you suggested required a thorough effort on our part. We believe that the manuscript has substantially improved paying attention to your suggestions.
- In the Title, ‘diagnostic’ should be replaced with ‘assay’.
Response:
Thank you for your advice,‘diagnostic’ has been replaced with ‘assay’.
- With respect to the emphasis of the paper that is now on the rather novel assay reagent, would it be better if the text on NT-proBNP (or preferably a shortened version of it) with which the Introduction starts (lines 33-44) was incorporated into the last paragraph of the Introduction?
Response:
Thank you for your advice, in the introduction part, the text on NT-proBNP was incorporated into lines 50-61.
